# Hydrogenotrophic methanogens of the mammalian gut: Functionally similar, thermodynamically different—A modelling approach

**Rafael Muñoz-Tamayo**[1]*, **Milka Popova**[2], **Maxence Tillier**[2], **Diego P. Morgavi**[2], **Jean-Pierre Morel**[3], **Gérard Fonty**[3], **Nicole Morel-Desrosiers**[3]

**1** UMR Modélisation Systémique Appliquée aux Ruminants, INRA, AgroParisTech, Université Paris-Saclay, Paris, France, **2** Institute National de la Recherche Agronomique, UMR1213 Herbivores, Clermont Université, VetAgro Sup, UMR Herbivores, Clermont-Ferrand, France, **3** Université Clermont Auvergne, CNRS, LMGE, Clermont-Ferrand, France

☯ These authors contributed equally to this work.
* rafael.munoz-tamayo@inra.fr

## Abstract

Methanogenic archaea occupy a functionally important niche in the gut microbial ecosystem of mammals. Our purpose was to quantitatively characterize the dynamics of methanogenesis by integrating microbiology, thermodynamics and mathematical modelling. For that, *in vitro* growth experiments were performed with pure cultures of key methanogens from the human and ruminant gut, namely *Methanobrevibacter smithii*, *Methanobrevibacter ruminantium* and *Methanobacterium formicium*. Microcalorimetric experiments were performed to quantify the methanogenesis heat flux. We constructed an energetic-based mathematical model of methanogenesis. Our model captured efficiently the dynamics of methanogenesis with average concordance correlation coefficients of 0.95 for $CO_2$, 0.98 for $H_2$ and 0.97 for $CH_4$. Together, experimental data and model enabled us to quantify metabolism kinetics and energetic patterns that were specific and distinct for each species despite their use of analogous methane-producing pathways. Then, we tested *in silico* the interactions between these methanogens under an *in vivo* simulation scenario using a theoretical modelling exercise. *In silico* simulations suggest that the classical competitive exclusion principle is inapplicable to gut ecosystems and that kinetic information alone cannot explain gut ecological aspects such as microbial coexistence. We suggest that ecological models of gut ecosystems require the integration of microbial kinetics with nonlinear behaviours related to spatial and temporal variations taking place in mammalian guts. Our work provides novel information on the thermodynamics and dynamics of methanogens. This understanding will be useful to construct new gut models with enhanced prediction capabilities and could have practical applications for promoting gut health in mammals and mitigating ruminant methane emissions.

**Data Availability Statement:** All relevant data are within the manuscript and its Supporting Information files.

**Funding:** This work received funding from Inra PHASE department and the Inra MEM metaprogramme to MP, RMT. The funder had no role in study design, data collection and analysis, decision to publish, or preparation of the manuscript.

**Competing interests:** The authors have declared that no competing interests exist.

## Introduction

Methanogenic archaea inhabit the gastro-intestinal tract of mammals where they have established syntrophic interactions within the microbial community [1–3] playing a critical role in the energy balance of the host [4,5]. In the human gut microbiota, the implication of methanogens in host homeostasis or diseases is poorly studied, but of growing interest [6]. *Methanobrevibacter smithii* (accounting for 94% of the methanogen population) and *Methanosphaera stadtmanae* are specifically recognized by the human innate immune system and contribute to the activation of the adaptive immune response [7]. Decreased abundance of *M. smithii* was reported in inflammatory bowel disease patients [8], and it has been suggested that methanogens may contribute to obesity [9]. In the rumen, the methanogens community is more diverse though still dominated by *Methanobrevibacter* spp., followed by *Methanomicrobium* spp., *Methanobacterium* spp. [10] and *Methanomassillicoccus* spp [11]. However, the proportion of these taxa could vary largely, with *Methanomicrobium mobile* and *Methanobacterium formicium* being reported as major methanogens in grazing cattle [12]. Though methanogens in the rumen are essential for the optimal functioning of the ecosystem (by providing final electron acceptors), the methane they produce is emitted by the host animal, contributing to global greenhouse gas (GHG) emissions. In the gastrointestinal tract of mammals, major rumen methanogens [13] and the dominant human archaeon *M. smithii* [14], are hydrogenotrophic archaea without cytochrome (membrane-associated electron transfer proteins). Cytochrome-lacking methanogens exhibit lower growth yields than archaea with cytochromes [15]. However, this apparent energetic disadvantage has been counterbalanced by a greater adaptation to the environmental conditions prevailing in the gastrointestinal tract [16], and by the establishment of syntrophic interactions with feed fermenting microbes. This syntrophic cooperation centred on hydrogen allows the anaerobic reactions of substrate conversion to proceed close to the thermodynamic equilibrium [17,18] (that is with Gibbs free energy change close to zero).

To our knowledge, thermodynamic considerations on human gut metabolism have been poorly addressed in existing mathematical models [19–22], although ttheoretical frameworks have been developed in other domains to calculate stoichiometric and energetic balances of microbial growth from the specification of the anabolic and catabolic reactions of microbial metabolism [23,24], and advances have been done to link thermodynamics to kinetics [25–27]. For the rumen, thermodynamic principles have been incorporated already into mathematical research frameworks because of their important role in feed fermentation. Thermodynamic studies have been performed to investigate theoretically (i) the profile of fermentation [28], (ii) alternative routes for hydrogen utilization [29], and (iii) the effect of hydrogen partial pressure on glucose fermentation and methanogenesis [30,31]. However, studies assessing quantitative comparisons between experimental data and model predictions using thermodynamic-based approaches have been of limited success in providing accurate predictions [32,33], probably due to missing controlling factors such as NADH oxidation and the dynamics of hydrogen partial pressure [31]. Another key factor explaining inaccurate predictions in existing rumen models is the lack of a dynamic representation of the microbial methanogens group. In this respect, new knowledge on the extent of methanogenesis and metabolic differences between the microbial members of this group could help to improve existing gut models. To our knowledge, none of the existing gut models integrates thermodynamics aspects to describe microbial growth of methanogens. Accordingly, our objective in this work was to develop a dynamic model with thermodynamic basis to quantify metabolism kinetics and energetic patterns that could inform on metabolic specificities between gut methanogenic archaea. Additionally, this model development aimed at providing tools for the analysis of ecological aspects (*e.g.*, competitive exclusion principle) of the methanogenic community that can be

instrumental when designing, for example, nutritional strategies for methane mitigation in ruminants. The model was built upon quantitative data characterizing the *in vitro* dynamics of hydrogen utilization, methane production, growth and heat flux of three hydrogenotrophic methanogenic species representing major human and ruminant genera: *Methanobrevibacter smithii*, *Methanobrevibacter ruminantium* and *Methanobacterium formicium*.

## Material and methods

### *In vitro* growth experiments

**Archaeal strains and growth media.** Archaeal strains used in the study were *M. ruminantium* M1 (DSM 1093), *M. smithii* PS (type strain DSM 861), and *M. formicium* MF (type strain DSM 1535). All archaeal strains were purchased from DSM previously. The strains were frozen at -80˚C, thawed and grown to get actively growing population before the beginning of the study. Balch growth media was prepared as previously described [34] and composition is summarized in S1 Table. Before autoclaving, 6 ml of media were distributed in Balch tubes (26 ml total volume) under $CO_2$ atmosphere.

**Experimental design and measures.** Starter cultures were grown until reaching optical density at 660 nm ($OD_{660}$) of 0.400 ± 0.030. Optical density was measured on a Jenway spectrophotometer (Bibby Scientific). Then, 0.6 ml were used to inoculate one experimental tube. Commercially prepared high purity $H_2/CO_2$ (80%/20%) gas mix was added to inoculated tubes by flushing for 1 min at 2.5 Pa. Mean initial $OD_{660}$ and pressure values are summarized in S2 Table. Growth kinetics for each strain were followed over 72 h. The experiment was repeated twice. Each kinetics study started with 40 tubes inoculated at the same time. At a given time point, two tubes with similar $OD_{660}$ values were sampled. The tubes were used for measuring gas parameters: pressure was measured using a manometer and composition of the gas phase was analysed by gas chromatography on a Micro GC 3000A (Agilent Technologies, France). The GC was equipped with two columns, MS-5A using argon as carrier gas and set to 100˚C and PPU using helium and set to 75˚C. The GC was calibrated using a certified gas standard mixture (Messer, France) containing methane, oxygen, hydrogen, carbon dioxide, and nitrogen. Approximately 2 ml of the sampled gas was injected in the GC for analysis. After gas sampling, one of the tubes was centrifuged 10 min at 13 000 g. The microbial pellet was weighed and stored at -20˚C in 2 ml screw-cap tubes containing 0.4 g of sterile zirconia beads (0.3 g of 1 mm and 0.1 g of 0.5 mm).

**DNA extraction and qPCR quantification of 16S rRNA genes.** One ml of lysis buffer (50mM NaCl, 50 mM TrisHCl pH 7.6, 50 mM EDTA, 5% SDS) was added directly to the frozen microbial pellet before homogenizing for $2 \times 30$ s at 5100 tours/min in a Precellys beadbeater (Bertin Instruments). Samples were centrifuged for 3 min at 14 000 g and the liquid phase transferred to a new tube before adding 600 µl of phenol–chloroform–3-methyl-1-butanol (25:24:1) solution. After centrifugation at 14 000 g for 3 min, the aqueous phase was transferred to a fresh tube and 500 µl of chloroform were added. The chloroform-washing step was repeated twice with centrifugation at 14000 g for 3 min between steps. The final volume of the aqueous phase was measured and DNA precipitation was initiated by adding 70% of the volume of isopropanol 100% and 10% of the volume of sodium acetate 3M. Sedimentation at 14 000 g for 30 min was again performed and the resulting DNA pellet was washed with 500 µl of ethanol 70% and dissolved in 50µl of molecular biology quality water. The extraction yield was checked on a Nanodrop 1000 Spectrophotometer (Thermo Fisher Scientific, France) and extracts run on a FlashGel System (Lonza, Rockland, Inc) to check integrity.

Copies of 16S rRNA genes were quantified using a qPCR approach. Primers used are those of Ohene-Adjei et al [35]; reaction assay and temperature cycles were as described previously

[36]. Triplicate qPCR quantification was performed on 20 ng of extracted DNA. Amplifications were carried out using SYBR Premix Ex Taq (TaKaRa Bio Inc., Otsu, Japan) on a StepOne system (Applied Biosystems, Courtabeuf, France). Absolute quantification involved the use of standard curves that had been prepared with gDNA of *Methanobrevibacter ruminantium* DSM 1093. PCR efficiency was of 103%. Results were expressed as copy numbers per ng of extracted DNA per g of microbial pellet. *M. smithii* and *M. ruminantium* strains used in this study possess two copies of 16S rRNA genes in their genomes. The number of cells was computed by dividing 16S copy numbers by 2.

## Microcalorimetry

Microcalorimetric experiments were performed to determine the heat flux pattern of each methanogen. Metabolic activity and microbial growth were monitored by using isothermal calorimeters of the heat-conduction type (A TAM III, TA Instruments, France) equipped with two multicalorimeters, each holding six independent minicalorimeters, allowed continuous and simultaneous recording as a function of time of the heat flux produced by 12 samples. The bath temperature was set at 39˚C; its long-term stability was better than $\pm 1 \times 10^{-4}$˚C over 24h. Each minicalorimeter was electrically calibrated. The specific disposable 4 mL microcalorimetric glass ampoules capped with butyl rubber stoppers and sealed with aluminium crimps were filled with 1.75 mL of Balch growth media and overpressed with 2.5 Pa of $H_2/CO_2$ 80%/20% gas mixture for 30 s. There was no significant difference in pressure at the beginning of the study. They were sterilized by autoclave and stored at 39˚C until the beginning of the microcalorimetric measurements. Actively growing cultures of methanogens ($OD_{660}$ of 0.280±0.030 for *M. smithii*, 0.271±0.078 for *M. ruminantium* and 0.142±0.042 for *M. formicium*) were stored at -20˚C in order to diminish microbial activity before inoculation. Cultures were thawed for 30 min at ambient temperature and inoculation was carried out by injecting 0.25 mL of the culture through the septum of the overpressed microcalorimetric ampoules just before inserting them into the minicalorimeters. Samples took about two hours to reach the bath temperature and yield a stable zero baseline. Blank experiments were also carried out by inserting ampoules that were not inoculated and, as expected, no heat flux was observed confirming the medium sterility. Each experiment was repeated thrice.

The heat flux $\left(\frac{dQ}{dt}\right)$, also called thermal power output $P$, was measured for each methanogen and blank samples with a precision $\geq 0.2$ µW. The heat flux data of each sample were collected every 5 minutes during more than 10 days. The total heat Q was obtained by integrating the overall heat flux time curve using the TAM Assistant Software and its integrating function (TA Instruments, France).

Classically, the heat flux-time curve for a growing culture starts like the S-shaped biomass curve (a lag phase followed by an exponential growth phase) but differs beyond the growth phase, the heat flux being then modulated by transition periods [37]. Heat flux data can be used to infer the microbial growth rate constant provided the existence of a correlation between isothermal microcalorimetry data and microbiological data (e.g., cell counts) at early growth [38]. During the exponential growth phase, microbial growth follows a first-order kinetics defined by the specific growth rate constant $\mu_c$ ($h^{-1}$). Analogously, the heat flux follows an exponential behaviour determined by the parameter $\mu_c$ as described by [37,38].

$$\frac{dQ}{dt} = \mu_c \cdot Q \qquad (1)$$

The growth rate constant $\mu_c$ can be determined by fitting the exponential part of the

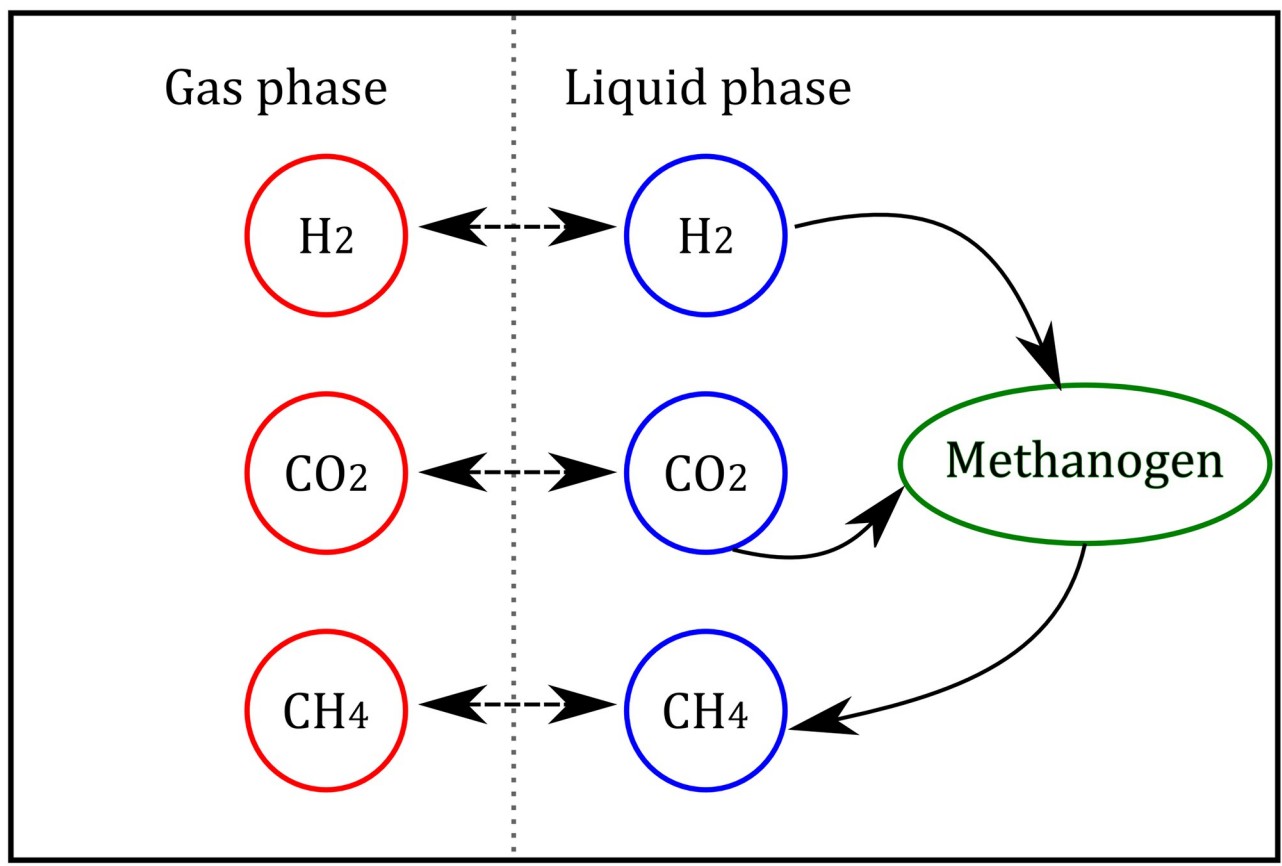

**Fig 1. Schematics of the *in vitro* methanogenesis process by hydrogenotrophic methanogens.** Double arrows represent fluxes due to liquid-gas transfer, simple arrows represent metabolic fluxes.

heat flux-time curve using the fitting function of the TAM Assistant Software. In our case study, careful selection of the exponential phase of heat flux dynamics was performed to provide a reliable estimation of the maximum growth rate constant from calorimetric data.

## Mathematical model development

**Modelling in vitro methanogenesis.** The process of *in vitro* methanogenesis is depicted in Fig 1. The $H_2/CO_2$ mixture in the gas phase diffuses to the liquid phase. The $H_2$ and $CO_2$ in the liquid phase are further utilized by the pure culture to produce $CH_4$. Methane in the liquid phase diffuses to the gas phase.

Model construction was inspired by our previous dynamic models of human gut [19] and rumen *in vitro* fermentation [39] followed by certain simplifications. The model considers the liquid-gas transfer of carbon dioxide. Due to the low solubility of hydrogen and methane [40], the concentration of these two gases in the liquid phase was not modelled. We assumed that the dynamics of concentrations in the gas phase are determined by kinetic rate of the methanogenesis. To incorporate thermodynamic information, instead of using the Monod equation in the original formulation, we used the kinetic rate function proposed by Desmond-Le Quéméner and Bouchez [26]. The resulting model is described by the following ordinary differential

equations

$$\frac{dx_{H_2}}{dt} = \mu_{max} \cdot \exp\left(-\frac{K_s \cdot V_g}{n_{g,H_2}}\right) \cdot x_{H_2} - k_d \cdot x_{H_2} \tag{2}$$

$$\frac{ds_{CO_2}}{dt} = -\frac{-Y_{CO_2} \cdot \mu_{max}}{Y} \cdot \exp\left(-\frac{K_s \cdot V_g}{n_{g,H_2}}\right) \cdot x_{H_2} - k_L a \cdot (s_{CO_2} - K_{H,CO_2} \cdot R \cdot T \cdot n_{g,CO_2}/V_g) \tag{3}$$

$$\frac{dn_{g,H_2}}{dt} = -\frac{\mu_{max}}{Y} \cdot \exp\left(-\frac{K_s \cdot V_g}{n_{g,H_2}}\right) \cdot V_L \cdot x_{H_2} \tag{4}$$

$$\frac{dn_{g,CO_2}}{dt} = V_L \cdot k_L a \cdot (s_{CO_2} - K_{H,CO_2} \cdot R \cdot T \cdot n_{g,CO_2}/V_g) \tag{5}$$

$$\frac{dn_{g,CH_4}}{dt} = \frac{Y_{CH_4} \cdot \mu_{max}}{Y} \cdot \exp\left(-\frac{K_s \cdot V_g}{n_{g,H_2}}\right) \cdot V_L \cdot x_{H_2} \tag{6}$$

where $s_{CO_2}$ is the concentration (mol/L) of carbon dioxide in the liquid phase and $x_{H_2}$ is the biomass concentration (mol/L) of hydrogenotrophic methanogens. The numbers of moles in the gas phase are represented by the variables $n_{g,H_2}$, $n_{g,CO_2}$, $n_{g,CH_4}$. The gas phase volume $V_g = 20$ mL and the liquid phase volume $V_L = 6$ mL. Liquid-gas transfer for carbon dioxide is described by a non-equilibria transfer rate which is driven by the gradient of the concentration of the gases in the liquid and gas phase. The transfer rate is determined by the mass transfer coefficient $k_L a$ ($h^{-1}$) and the Henry's law coefficients $K_{H,CO_2}$ (M/bar). $R$ (bar·(M · K)$^{-1}$) is the ideal gas law constant and $T$ is the temperature (K). Microbial decay is represented by a first-order kinetic rate with $k_d$ ($h^{-1}$) the death cell rate constant. Microbial growth was represented by the rate function proposed by Desmond-Le Quéméner and Bouchez [26] using hydrogen as single substrate

$$\mu = \mu_{max} \cdot \exp\left(-\frac{K_s \cdot V_g}{n_{g,H_2}}\right) \tag{7}$$

where $\mu$ is the growth rate ($h^{-1}$), $\mu_{max}$ ($h^{-1}$) is the maximum specific growth rate constant and $K_s$(mol/L) the affinity constant. Eq (7) is derived from energetic principles following Boltzmann statistics and uses the concept of exergy (maximum work available for a microorganism during a chemical transformation). The affinity constant has an energetic interpretation since it is defined as

$$K_s = \frac{E_{M} + E_{dis}}{v_{harv} \cdot E_{cat}} \tag{8}$$

where $E_{dis}$ (kJ/mol) and $E_M$ (kJ/mol) are the dissipated exergy and stored exergy during growth respectively. $E_{cat}$ (kJ/mol) is the catabolic exergy of one molecule of energy-limiting substrate, and $v_{harv}$ is the volume at which the microbe can harvest the chemical energy in the form of substrate molecules [26]. $E_{cat}$ is the absolute value of the Gibbs energy of catabolism ($\Delta G_{r,c}$) when the reaction is exergonic ($\Delta G_{r,c} < 0$) or zero otherwise. The stored exergy $E_M$ is calculated from a reaction (destock) representing the situation where the microbe gets the energy by consuming its own biomass. $E_M$ is the absolute value of the Gibbs energy of biomass consuming

reaction ($\Delta G_{r,destock}$) when the reaction is exergonic ($\Delta G_{r,destock} < 0$) or zero otherwise. Finally, the dissipated exergy $E_{dis}$ is the opposite of the Gibbs energy of the overall metabolic reaction, which is a linear combination of the catabolic and destock reactions. This calculation follows the Gibbs energy dissipation detailed in Kleerebezem and Van Loosdrecht [24].

In our model, the stoichiometry of methanogenesis is represented macroscopically by one catabolic reaction ($R_1$) for methane production and one anabolic reaction ($R_2$) for microbial formation. We assumed that ammonia is the only nitrogen source for microbial formation. The molecular formula of microbial biomass was assumed to be $C_5H_7O_2N$ [40].

$$R_1 : 4\ H_2 +\ CO_2 \rightarrow CH_4 + 2\ H_2O$$

$$R_2 : 10\ H_2 +\ 5\ CO_2 + NH_3 \rightarrow C_5H_7O_2N +\ 8\ H_2O$$

In the model, the stoichiometry of the reactions is taken into account *via* the parameters $Y$, $Y_{CO_2}$, $Y_{CH_4}$, which are the yield factors (mol/mol) of microbial biomass, $CO_2$ and $CH_4$. The microbial yield factor $Y$ was extracted from literature. We assumed that *M. smithii* and *M. ruminantium* have the same yield (being both Methanobrevibacter). This yield factor was set to 0.006 mol biomass/mol $H_2$, using the methane-based molar growth yield of 2.8 g biomass/ mol $CH_4$ estimated for *M. smithii* [41] and the Eqs (9) and (11). Similarly, the yield factor for *M. formicium* was set to 0.007 mol biomass/mol $H_2$ using the methane-based molar growth yield of 3.5 g biomass/mol $CH_4$ reported by Schauer and Ferry [42]. The fraction of $H_2$ utilized for microbial growth (reaction $R_2$) is defined by the yield factor $Y$ (mol of microbial biomass/ mol of $H_2$). Now, let $f$ be the fraction of $H_2$ used for the catabolic reaction $R_1$. Reaction $R_2$ tells us that for every 10 moles of $H_2$ used in $R_2$, we get 1 mol of microbial biomass. Hence, it follows that

$$Y = \frac{1}{10} \cdot (1 - f) \tag{9}$$

If $Y$ is known, the fraction $f$ can be computed from Eq (9).

The yield factors of $CO_2$ and $CH_4$ can be expressed as functions of the the fraction $f$:

$$Y_{CO_2} = \left(\frac{1}{4}\right) \cdot f + \left(\frac{5}{10}\right) \cdot (1 - f) \tag{10}$$

$$Y_{CH_4} = \left(\frac{1}{4}\right) \cdot f \tag{11}$$

The model has two physicochemical parameters ($k_La$, $K_{H,CO_2}$) and four biological parameters ($\mu_{max}$, $K_s$, $Y$, $k_d$). The initial condition for $s_{CO_2}$ is unknown and was also included in the parameter vector for estimation. The Henry's law coefficients are known values calculated at 39°C using the equations provided by Batstone et al. [40]. An implementation of the model in the Open Source software Scilab (https://www.scilab.org/) is available at https://doi.org/10.5281/zenodo.3271611.

**Theoretical model to study interactions among methanogens.** The experimental study of microbial interactions requires sophisticated *in vitro* systems under continuous operation such as the one developed by Haydock *et al.* [43]. In our work, we explored by means of mathematical modelling how the methanogens can interact under *in vivo* conditions. For this theoretical study, we elaborated a toy model based on the previous model for *in vitro* methanogenesis. Let us consider the following simple model for representing the consumption of

hydrogen by the methanogenic species $i$ under an *in vivo* scenario of continuous flow

$$\frac{dx_{\mathrm{H}_2,i}}{dt} = \mu_{\mathrm{max},i} \cdot \exp\left(-\frac{K_{\mathrm{s},i} \cdot V_{\mathrm{g}}}{n_{\mathrm{g,H}_2}}\right) \cdot x_{\mathrm{H}_2;i} - D_i \cdot x_{\mathrm{H}_2,i} \tag{12}$$

$$\frac{dn_{\mathrm{g,H}_2}}{dt} = q_{\mathrm{H}_2} - \frac{\mu_{\mathrm{max},i}}{Y_i} \cdot \exp\left(-\frac{K_{\mathrm{s},i} \cdot V_{\mathrm{g}}}{n_{\mathrm{g,H}_2}}\right) \cdot V_{\mathrm{L}} \cdot x_{\mathrm{H}_2,i} - b \cdot n_{\mathrm{g,H}_2} \tag{13}$$

where $q_{\mathrm{H}_2}$ (mol/h) is the flux of hydrogen produced from the fermentation of carbohydrates. The kinetic parameters are specific to the species $i$ ($x_{\mathrm{H}_2,i}$). The parameter $D_i$ (h$^{-1}$) is the dilution rate of the methanogens and $b$ (h$^{-1}$) is an output substrate rate constant. Extending the model to $n$ species with a common yield factor $Y$, the dynamics of hydrogen is given by

$$\frac{dn_{\mathrm{g,H}_2}}{dt} = q_{\mathrm{H}_2} - \frac{V_{\mathrm{L}}}{Y} \sum_{i=1}^{n} \mu_{\mathrm{max},i} \cdot \exp\left(-\frac{K_{\mathrm{s},i} \cdot V_{\mathrm{g}}}{n_{\mathrm{g,H}_2}}\right) \cdot x_{\mathrm{H}_2,i} - b \cdot n_{\mathrm{g,H}_2} \tag{14}$$

where the sub index $i$ indicates the species. In our case study, $n = 3$.

## Parameter identification

Before performing the numerical estimation of the model parameters, we addressed the question of whether it was theoretically possible to determine uniquely the model parameters given the available measurements from the experimental setup. This question is referred to as structural identifiability [44]. Structural identifiability analysis is of particular relevance for model whose parameters are biologically meaningful, since knowing the actual value of the parameter is useful for providing biological insight on the system under study [45]. Moreover, in our case, we are interested in finding accurate estimates that can be further used as priors in an extended model describing the *in vivo* system.

We used the freely available software DAISY [46] to assess the structural identifiability of our model. Physical parameters ($k_{\mathrm{L}}a$, $K_{\mathrm{H,CO}_2}$) were set to be known. The model was found to be structurally globally identifiable. In practice, however, to facilitate the actual identification of parameters and reduce practical identifiability problems such as high correlation between the parameters [47], we fixed some model parameters to values reported in the literature. The transport coefficient $k_{\mathrm{L}}a$, the Henry's law coefficient $K_{\mathrm{H,CO}_2}$, and the dead cell rate constant $k_{\mathrm{d}}$ were set to be known and were extracted from Batstone et al. [40]. Therefore, only the parameters $\mu_{\mathrm{max}}$, $K_{\mathrm{s}}$ and initial condition of $s_{\mathrm{CO}_2}$ were set to be estimated. To capitalize on the calorimetric data, we further assumed that $\mu_{\mathrm{max}}$ was equal to the specific rate constant $\mu_{\mathrm{c}}$ estimated from the heat flux-time curve. By this, only the affinity constant for each strain and the initial condition of $s_{\mathrm{CO}_2}$ were left to be estimated.

The parameter identification for each methanogen was performed with the IDEAS Matlab$^{\circledR}$ toolbox [48] (freely available at http://genome.jouy.inra.fr/logiciels/IDEAS). The parameter identification was performed with the data of the *in vitro* growth experiments. The measured variables are the number of moles in the gas phase (H$_2$, CH$_4$, CO$_2$). The Lin's concordance correlation coefficient (CCC) [49] was computed to quantify the agreement between the observations and model predictions.

## Results

### Methanogens biomass

Archaea-specific primers targeting the 16S rRNA gene were used to enumerate microbial cells in each pure culture. Three hours post inoculation microbial numbers varied from $7.62\times10^7$ to $2.81\times10^8$ and reached $10^9$ after 72 hours of incubation. S3 Table summarizes microbial numbers at different sampling times.

### Calorimetric pattern of methanogens

Fig 2 displays a representative isothermal calorimetric curve for each methanogen. The three measured heat flux dynamics of each methanogen were found to follow similar energetic patterns. *M. smithii* and *M. formicium* exhibited a lag phase of a few hours, while *M. ruminantium* was already metabolically active when introduced into the minicalorimeter though several attempts were made to obtain a lag phase by changing storage conditions and thawing the

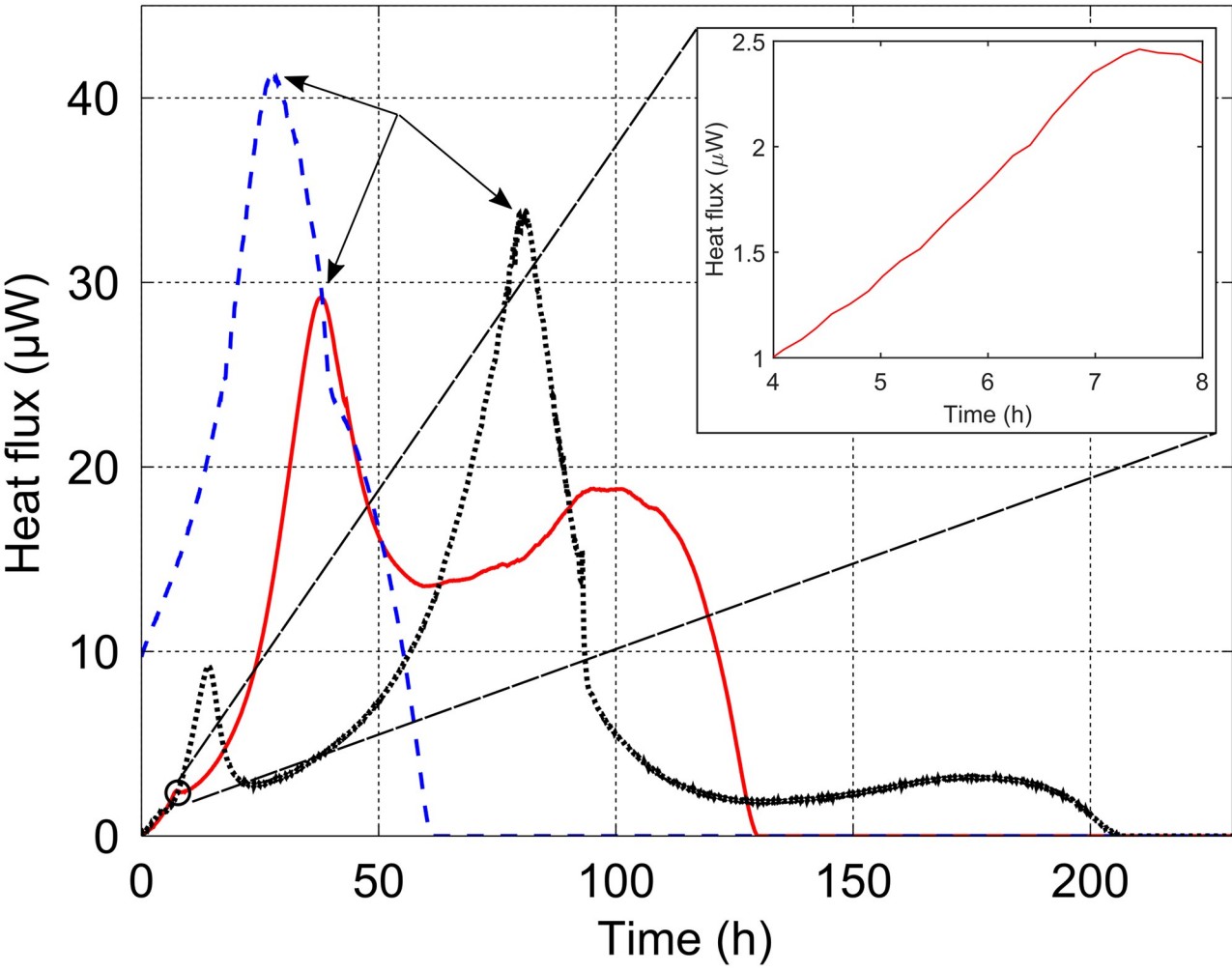

**Fig 2. Example of isothermal calorimetric curves for *M. ruminantium* (dashed blue line), *M. smithii* (solid red line) and *M. formicium* (dotted black line).** The dominant metabolic phase is represented by one peak (shown with the arrows). The magnitude of the peak differs between the methanogens and also the slope of the heat flux trajectories. The return of the heat flux to the zero baseline also differs between the three methanogens. The inset zoom displays the peak exhibited by *M. smithii* at 7.4 h.

culture just before inoculating the microcalorimetric ampoules. The pattern of heat flux for all tested methanogens is characterized by one predominant peak which was observed at different times for each methanogen. *M. smithii* exhibited a second metabolic event occurring at 60 h with an increase of heat flux. The same phenomenon was observed for *M. formicium* but at a lower intensity that started at 140 h. The process was considered completed when the heat flux ceased marking the end of the metabolic activity. It is noted that *M. formicium* produced a small peak at 14 h (Fig 2). A similar peak, but of much smaller size, was observed on the other curves obtained with this methanogen. *M. smithii* also exhibits a small peak (occurrence of 3 out of 3) at 7.4 h shown in the inset of Fig 2.

The total heat ($Q_m$) produced during the methanogenesis process that took place under the present experimental conditions was, on average, -5.5 ± 0.5 J for the three methanogens (for *M. ruminantium*, the missing initial part of the heat flux-time curve was approximately estimated by extrapolating the exponential fit). As we shall see below, this experimental value is consistent with the theoretically expected value.

## Estimation of thermodynamic properties

We defined two macroscopic reactions to represent the catabolism (R1) and anabolism (R2) of the methanogenesis (see Modelling in vitro methanogenesis section). All thermodynamic properties result from the contribution of both catabolic and anabolic reactions. The calculations of total heat ($Q_m$), enthalpy ($\Delta H_m$), Gibbs energy ($\Delta G_m$) and entropy ($\Delta S_m$) of the methanogenesis are detailed in S4 Table. The estimated overall heat produced during the methanogenesis process under our experimental conditions was in average $Q_m = -5.66$J. This heat results from the sum of the heat of the catabolic reaction ($Q_c$) and the heat of the anabolic reaction ($Q_a$). From the total heat of the methanogenesis, the anabolic reaction contributes to 7% of the metabolic heat for *M. smithii* and *M. ruminantium*. For *M. formicium*, the contribution of the anabolic reaction to the metabolic heat is 9%. It is also interesting to note that there is a very good agreement between the theoretical value calculated above and the overall heat experimentally determined by microcalorimetry (-5.5 ± 0.5 J).

Table 1 shows the thermodynamic properties per mole of biomass formed during methanogenesis of *M. ruminantium*, *M. smithii* and *M. formicium* on $H_2/CO_2$. These properties are compared with values found in the literature for other methanogens grown on different substrates.

## Dynamic description of *in vitro* kinetics

The developed mathematical model was calibrated with the experimental data from *in vitro* growth experiments in Balch tubes. Table 2 shows the parameters of the dynamic kinetic model described in Eqs 2–6. The reported value of $\mu_{max}$ for each methanogen corresponds to

**Table 1. Gibbs energies, enthalpies and entropies of metabolic processes involving some methanogens growing on different energy sources.**

| Microorganism | Energy substrate | Growth conditions | $\Delta G_m$ kJ / C-mol | $\Delta H_m$ kJ / C-mol | $T\Delta S_m$ kJ / K⁻¹ C-mol | Driving force | Reference |
|---|---|---|---|---|---|---|---|
| *M. ruminantium, M. smithii,* | $H_2/CO_2$ | anaerobic | -1073 | -2132 | -1059 | Enthalpy-driven but Entropy-retarded | this work |
| *M. formicium* | $H_2/CO_2$ | anaerobic | -801 | -1605 | -804 | Enthalpy-driven but Entropy-retarded | this work |
| *M. thermo-autotrophicum* | $H_2/CO_2$ | anaerobic | -802 | -3730 | -2928 | Enthalpy-driven but Entropy-retarded | [50] |
| *M. formicium* | formate | anaerobic | -880 | -613 | +267 | Enthalpy-driven | [23] |
| *M. barkeri* | methanol | anaerobic | -570 | -420 | +150 | Enthalpy-driven | [23] |
| *M. barkeri* | acetate | anaerobic | -366 | +145 | +511 | Entropy-driven but enthalpy-retarded | [51] |

**Table 2. Parameters of the model of *in vitro* methanogenesis.** The value reported $\mu_{max}$ for each methanogen is the mean value obtained from three heat flux-time curves.

| Parameter | Definition | Value | | |
|---|---|---|---|---|
| $k_L a$ (h$^{-1}$) | Liquid–gas transfer constant | 8.33 | | |
| $K_{H,CO_2}$ (M/bar) | Henry's law coefficient of carbon dioxide | 0.0246 | | |
| $k_d$ (h$^{-1}$) | Death cell rate constant | 8.33x10$^{-4}$ | | |
| | | *M. smithii* | *M. ruminantium* | *M. formicium* |
| $K_s$ (mol/L) | Affinity constant | 0.028 | 0.042 | 0.011 |
| $\mu_{max}$ (h$^{-1}$) | Maximum specific growth rate constant | 0.12 | 0.07 | 0.046 |
| $Y$ (mol biomass /mol H$_2$) | Microbial biomass yield factor | 0.006 | 0.006 | 0.007 |

the average value obtained from three heat flux-time curves. From Table 2, it is concluded that *M. smithii* exhibited the highest growth rate constant, followed by *M. ruminantium* and finally *M. formicium*. In terms of the affinity constant $K_s$, while *M. smithii* and *M. ruminantium* are of the same order, the affinity constant for *M. formicium* is lower in one order of magnitude.

Fig 3 displays the dynamics of the compounds in the methanogenesis for the three methanogens. The data of the figure is available at https://doi.org/10.5281/zenodo.3469655. Experimental data are compared against the model responses. Table 3 shows standard statistics for model evaluation. The model captures efficiently the overall dynamics of the methanogenesis. Hydrogen and methane are very well described by the model with average concordance correlation coefficients (CCC) of 0.98 and 0.97 respectively. For carbon dioxide, CCC = 0.95.

Fig 4 displays the dynamics for the methanogens as measured by the 16S rRNA gene, as well as the dynamics of biomass as predicted for the model. As observed, the microbes follow a typical Monod-like trajectory.

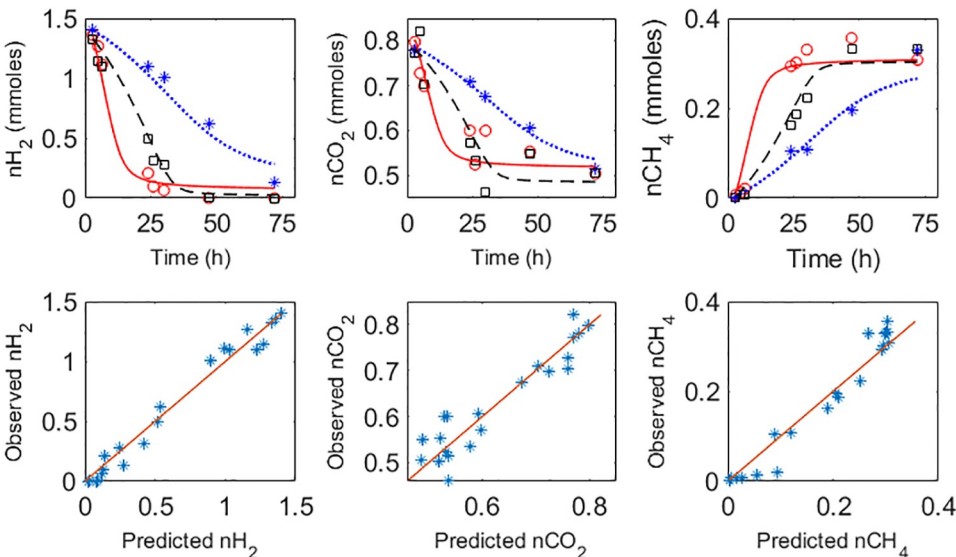

**Fig 3. Assessment of model performance.** Top plots: dynamics of methanogenesis by *M. ruminantium* (*), *M. smithii* (○) and *M. formicium* (□). Experimental data (*,○,□) are compared against model predicted responses: dotted blue lines (*M. ruminantium*), solid red lines (*M. smithii*) and dashed black lines (*M. formicium*). Bottom plots: summary observed vs predicted variables. The solid red line is the isocline.

**Table 3. Statistical indicators for model evaluation.**

| | Hydrogen | | | Methane | | | Carbon dioxide | | |
|---|---|---|---|---|---|---|---|---|---|
| | CCC* | $r^2$ | CV$_{RMSE}$** | CCC* | $r^2$ | CV$_{RMSE}$** | CCC* | $r^2$ | CV$_{RMSE}$** |
| *M. smithii* | 0.99 | 0.98 | 16 | 0.96 | 0.94 | 18 | 0.93 | 0.84 | 6 |
| *M. ruminantium* | 0.97 | 0.95 | 11 | 0.96 | 0.93 | 20 | 0.99 | 0.98 | 2 |
| *M. formicium* | 0.99 | 0.98 | 13 | 0.98 | 0.96 | 17 | 0.92 | 0.86 | 8 |
| Mean | 0.98 | 0.97 | 13 | 0.97 | 0.94 | 18 | 0.95 | 0.89 | 5 |

* CCC: Lin's concordance correlation coefficient.

** CV$_{RMSE}$: coefficient of variation of the root mean squared error.

## Discussion

Our objective in this work was to quantitatively characterize the dynamics of hydrogen utilization, methane production, growth and heat flux of three hydrogenotrophic methanogens by integrating microbiology, thermodynamics and mathematical modelling. Our model developments were instrumental to quantify energetic and kinetic differences between the three methanogens studied, strengthening the potentiality of microcalorimetry as a tool for characterizing the metabolism of microorganisms [52]. This modelling work provides estimated parameters that can be used as prior values for other modelling developments of gut microbiota.

### Energetic and kinetic differences between methanogens

Methanogenesis appears as a simple reaction described by a single substrate kinetic rate on $H_2$. The microcalorimetry approach we applied revealed that this simplicity is only apparent and that hydrogenotrophic methanogens exhibit energetic and kinetic differences. Methanogenesis is indeed a complex process that can be broken down in several stages. The dominant metabolic phase is represented by one peak that occurs at different times. The magnitude of the peak differs between the methanogens and also the slope of the heat flux trajectories. For *M. smithii* and *M. formicium* the main peak was preceded by a small increase in heat flux which translates in a metabolic activity that remains to be elucidated. For *M. ruminantium*, we do not

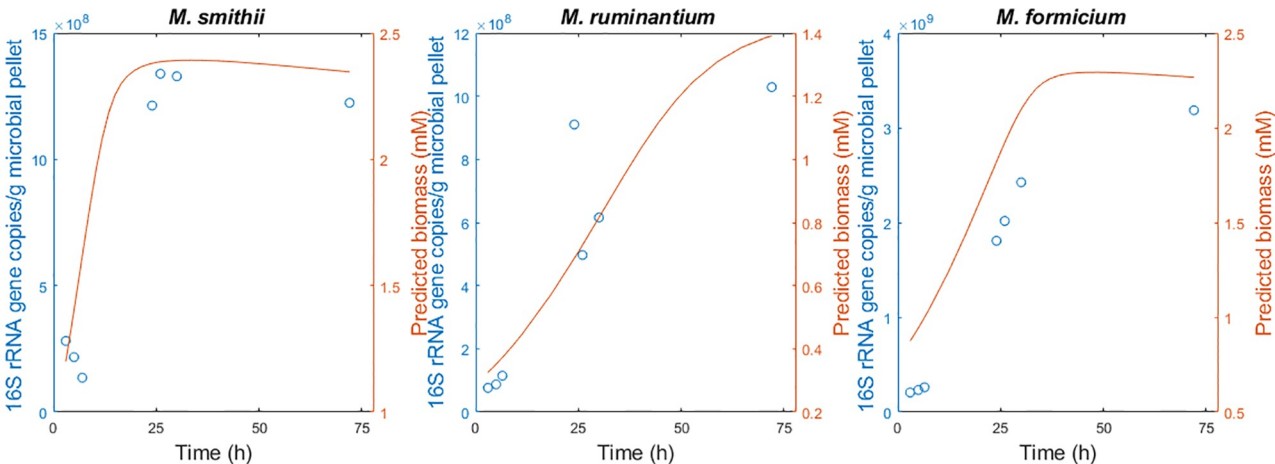

**Fig 4. Dynamics of methanogens as measured by 16S rRNA gene copies (circles) and biomass concentrations (solid line) predicted by the model.**

know whether the small peak exists since the initial part of the curve is missing. For *M. smithii* and *M. formicium*, it was observed a metabolic event after the main peak around 60 h for *M. smithii* and 140h *M. formicium*. This peak could be representing cell lysis process [38]. The return time of the heat flux to the zero baseline was also different. The energetic difference is associated with kinetic differences that translate into specific kinetic parameters, namely affinity constant ($K_s$) and maximum growth rate constant ($\mu_{max}$). Previously, energetic differences between methanogens have been ascribed to the presence or absence of cytochromes [15]. These differences are translated into different yield factors, $H_2$ thresholds, and doubling times. The kinetic differences revealed in this study for three cytochrome-lacking methanogens indicate that factors other than the presence of cytochromes might play a role in the energetics of methanogenesis. Interestingly, calorimetric experiments showed that *M. ruminantium* was metabolically active faster than the other methanogens, characteristic that could explain the predominance of *M. ruminantium* in the rumen [53]. Looking at the expression of the affinity constant (Eq (8)), the differences between the affinity constants among the methanogens can be explained by the differences between the by the harvest volume $v_{harv}$ and the yield factors. Note that in the kinetic function developed by Desmond-Le Quéméner and Bouchez [26], the maximum growth rate did not have any dependency on the energetics of the reaction. Our experimental study revealed that $\mu_{max}$ is species-specific and reflects the dynamics of the heat flux of the reaction at the exponential phase. This finding suggests that a further extension of the kinetic model developed by Desmond-Le Quéméner and Bouchez [26] should include the impact of energetics on $\mu_{max}$. Since our study is limited to three species, it is important to conduct further research on other methanogens to validate our findings. In this same line, to enhance the evaluation of the predictive capabilities of our model, a further model validation is required with independent data set under different experimental conditions (*e.g.* continuous mode operation) to those used in this study.

## Energetic analysis

Regarding the energetic information for different methanogens summarized in Table 1, it is observed that the thermodynamic behaviour of the three methanogens is analogous to that observed for *Methanobacterium thermoautotrophicum* [50]. The values reported in Table 1 show indeed that the methanogenesis on $H_2/CO_2$ is characterized by large heat production. The growth is highly exothermic, with a $\Delta H_m$ value that largely exceeds the values found when other energy substrates are used. The enthalpy change $\Delta H_m$, which is more negative than the Gibbs energy change $\Delta G_m$, largely controls the process. Growth on $H_2/CO_2$ is also characterized by a negative entropic contribution $T\Delta S_m$ which, at first sight, may look surprising since entropy increases in most cases of anaerobic growth [54]. However, this can be understood if one remembers that $T\Delta S_m$ corresponds in fact to the balance between the final state and the initial state of the process, that is

$$T\Delta S_m = \frac{(1-10Y)}{4Y}\ T\Delta S_c + T\Delta S_a = \frac{(1-10Y)}{4Y}\ T(S_{final}-S_{initial})_c + T(S_{final}-S_{initial})_a$$

Methanogenesis on $H_2/CO_2$ is particular because the final state of its catabolic reaction (1 mol $CH_4$ + 2 mol $H_2O$) involves a smaller number of moles than the initial state (4 mol $H_2$ + 1 mol $CO_2$), which results in a significant loss of entropy during the process. For spontaneous growth in such a case, the $\Delta H_m$ must not only contribute to the driving force but must also compensate the growth-unfavourable $T\Delta S_m$, which means that $\Delta H_m$ must be much more negative than $\Delta G_m$ [55]. For this reason, methanogenesis on $H_2/CO_2$, which is accompanied by a considerable decrease of entropy and a large production of heat, has been designed as an

entropy-retarded process [50]. More generally, von Stockar and Liu [55] noticed that when the Gibbs energy of the metabolic process is resolved into its enthalpic and entropic contributions, very different thermodynamic behaviours are observed depending on the growth type. These thermodynamic behaviours are: aerobic respiration is clearly enthalpy-driven ($\Delta H_m \ll 0$ and $T\Delta S_m > 0$), whereas fermentative metabolism is mainly entropy-driven ($\Delta H_m < 0$ and $T\Delta S_m \gg 0$). Methanogenesis on $H_2/CO_2$ is enthalpy-driven but entropy-retarded ($\Delta H_m \ll 0$ and $T\Delta S_m < 0$), whereas methanogenesis on acetate is entropy-driven but enthalpy-retarded ($\Delta H_m > 0$ and $T\Delta S_m \gg 0$). In the present case, the highly exothermic growth of *M. ruminantium*, *M. smithii* and *M. formicium* on $H_2/CO_2$ is largely due to the considerable decrease of entropy during the process: in fact, 50% of the heat produced here serves only to compensate the loss of entropy. A proportion of 80% was found for *M. thermoautotrophicum* [50], which results from the fact that their $T\Delta S_m$ and $\Delta H_m$ values are, respectively, 2.7 and 1.7 times larger than ours. This difference might be due to the differences in the temperature of the studies, namely 39°C in our study vs 60°C in the study by Schill et al. [50].

## Do our results inform on ecological questions such as species coexistence?

The competitive exclusion principle [56] states that coexistence cannot occur between species that occupy the same niche (the same function). Only the most competitive species will survive. Recently, by using thermodynamic principles, Großkopf & Soyer [27] demonstrated theoretically that species utilizing the same substrate and producing different compounds can coexist by the action of thermodynamic driving forces. Since in our study the three methanogens perform the same metabolic reactions, the thermodynamic framework developed Großkopf & Soyer [27] predicts, as the original exclusion principle [56], the survival of only one species. By incorporating thermodynamic control on microbial growth kinetics, Lynch et al [57] showed theoretically that differentiation of ATP yields can explain ecological differentiation of methanogens over a range of liquid turnover rates. This theoretical work predicts that for a fixed liquid turnover rate, only one species survives. For the continuous culture of microorganisms, it has been demonstrated that at the equilibrium (growth rate equals the dilution rate) with constant dilution rates and substrate input rates, the species that has the lowest limiting substrate concentration wins the competition. From Eq (12), the number of moles of hydrogen of the species $n^*_{g,H_2,i}$ at the steady state is

$$n^*_{g,H_2,i} = \frac{K_{s,i} \cdot V_g}{\log(\mu_{max,i}/D_i)}$$

Using the model parameters of Table 2, we studied *in silico* three possible competition scenarios, assuming a constant environment (constant dilution rate *D*). Two dilution rates were evaluated: $D = 0.021$ h$^{-1}$ (retention time = 48 h) and $D = 0.04$ h$^{-1}$ (retention time = 25 h). A retention time of 48 h corresponds to values measured in small ruminants [58] and to humans as we used in our gut model [19]. For higher retention times, the results obtained for 48 h hold. For $D = 0.021$ h$^{-1}$, we obtained that $n^*_{g,H_2,Ms} = 0.32$ mmol, $n^*_{g,H_2,Mr} = 0.68$ mmol, $n^*_{g,H_2,Mf} = 0.28$ mmol, where the subindex Ms, Mr, Mf stand for *M. smithii*, *M. ruminantium* and *M.formicium*. From these results, it appears that under a constant environment, *M. formicium* will win the competition. Since $n^*_{g,H_2,Ms} < n^*_{g,H_2,Mr}$, *M. ruminantium* will be extinguished before *M. smithii*. For $D = 0.04$ h$^{-1}$, we obtained that $n^*_{g,H_2,Ms} = 0.49$ mmol, $n^*_{g,H_2,Mr} = 1.42$ mmol, $n^*_{g,H_2,Mf} = 1.57$ mmol, and thus *M. smithii* wins the competition. To win the competition, *M. ruminantium* requires longer retention times than its competitors. Retention times of digesta longer than 48 h are physiologically uncommon, thus the presence

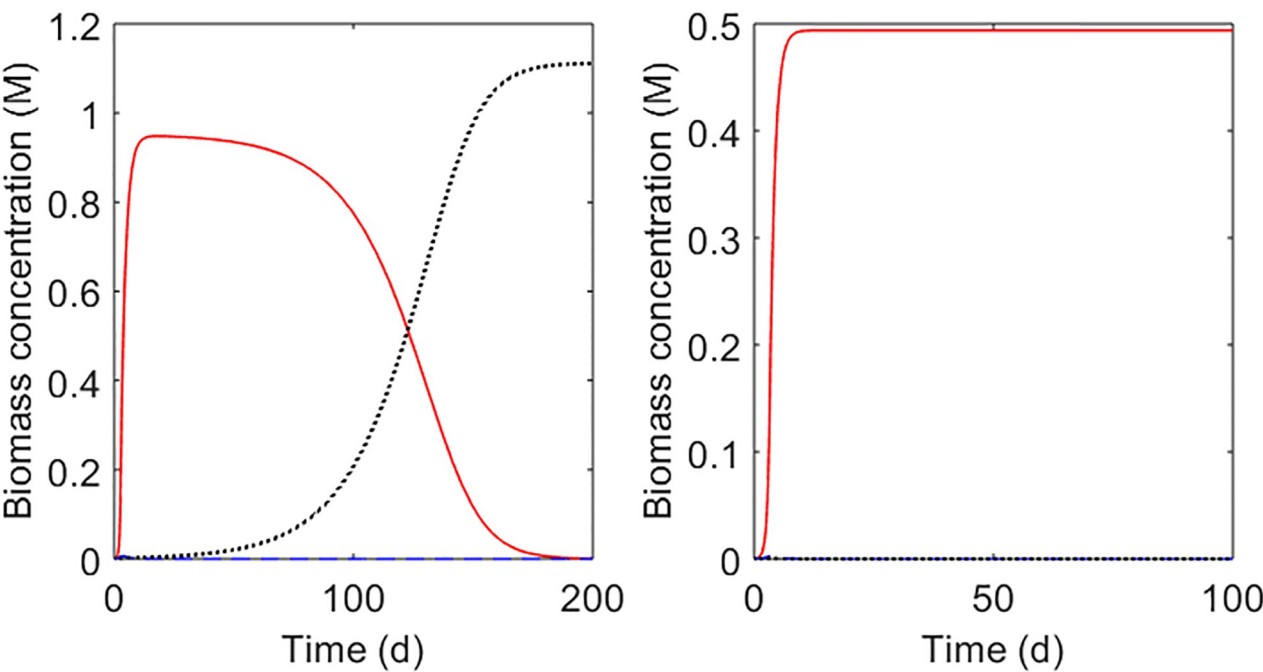

**Fig 5. Possible competition scenarios between *M. ruminantium* (blue dashed line), *M. smithii* (red solid line) and *M. formicium* (black dotted line) in a hypothetical constant environment.** A. At constant dilution rate of 0.021 h$^{-1}$, *M. formicium* displaces the other two methanogens. B. With a constant dilution rate of 0.04 h$^{-1}$, *M. smithii* wins the competition. At constant environmental conditions, only one species wins and displaces the other methanogens.

of *M. ruminantium* in the gut ecosystem can be explained, for example, from known adhesion properties (both *M. ruminantium* and *M. smithii* genes encode adhesin-like proteins [59,60]. To illustrate these aspects, we built a multiple-species model with the three methanogens using Eqs (12) and (14). The parameter *b* was set to 0.5 h$^{-1}$ and the hydrogen flux production $q_{H_2}$ rate was set to 0.02 mol/min. Fig 5A displays the dynamics of the three methanogens for the first scenario (*D* = 0.021 h$^{-1}$). It is observed that at 50 d only *M. formicium* survives. This result, however, is not representative of what occurs in the rumen where the three methanogens coexist [5,61]. It is intriguing that in our toy model it is *M. formicium* that wins the competition, bearing in mind that *M. ruminantium* and *M. smithii* are more abundant than *M. formicium* [5,53]. Fig 5 shows that selective conditions favour the survival of one species. Similar results can be obtained for the human gut by including the effect of pH on microbial growth [22] and setting the gut pH to select one of the species.

On the basis of the competitive exclusion principle, it is thus intriguing that having a very specialized function, methanogens are a diverse group that coexist. Gut ecosystems, therefore, exhibit the paradox of the plankton introduced by Hutchinson (1961) that presents the coexistence of species all competing for the same substrate in a relatively isotropic or unstructured environment [62]. In the case of the rumen, our modelling work suggests that in addition to kinetic and thermodynamic factors, other forces contribute to the ecological shaping of the methanogens community in the rumen favouring the microbial diversity. Indeed, methanogenic diversity in the rumen results from multiple factors that include pH sensitivity, the association with rumen fractions (fluid and particulate material), and the endosymbiosis with rumen protozoa [5,53]. For the human gut, ecological factors enable methanogens to coexist to a competitive environment where hydrogenotrophic microbes (acetogens, methanogenic

archaea and sulfate-reducing bacteria) utilize $H_2$ *via* different pathways [63–65]. Both in the human gut and in the rumen, microbes grow in association with biofilms that form a polymer-based matrix that provides nutritional and hydraulic advantages for microbial growth and resistance to shear forces [19,66]. Indeed, in our modelling work of human gut fermentation [19], we suggested that, from the different actions the mucus has on colonic fermentation, the mechanism of promoting conditions for microbial aggregation appears as the most relevant factor for attaining the high microbial density and the high level of fibre degradation characteristic of the human gut. Altogether, these factors result in nonlinear behaviours, spatial and temporal variations that promote coexistence and diversity, that, as discussed in dedicated literature on microbial ecology [67–71], render the classical formulation of the competitive exclusion principle [56,72] inapplicable to gut ecosystems.

Finally, mathematical modelling is expected to enhance our understanding of gut ecosystems [66,73]. It is then key that in addition to metabolic aspects, mathematical models of gut fermentation incorporate the multiple aspects that shape microbial dynamics to provide accurate predictions and improve insight on gut metabolism dynamics and its potential modulation. For ruminants, the development of precision livestock technologies provides promising alternatives for integrating real-time data of key animal phenotypes such as feeding behaviour with mathematical models for estimating methane emissions [74] and rumen function indicators at large scale. These tools will be instrumental to support livestock management decisions and guide timely interventions. Similarly, for humans, mathematical models coupled with electronic technologies for online monitoring of gut function [75] might facilitate the diagnosis and the design of personalized therapies for gastrointestinal diseases.

## Supporting information

**S1 Table. Methanogens growth media composition.**
(DOCX)

**S2 Table. Summary of initial OD and pressure measured immediately after primary inoculation.**
(DOCX)

**S3 Table. qPCR quantification of 16S rRNA genes.**
(DOCX)

**S4 Table. Calculation of thermodynamic properties of the methanogenesis.**
(DOCX)

## Acknowledgments

We are grateful to Dominique Graviou (UMRH, Inra) for her skilled assistance on the *in vitro* growth experiments and qPCR assays. We thank the Inra PHASE department and the Inra MEM metaprogramme for financial support. RMT, MP and DPM acknowledge the support of ERA-net gas co-fund for funding the RumenPredict project.

## Author Contributions

**Conceptualization:** Rafael Muñoz-Tamayo, Milka Popova, Diego P. Morgavi, Nicole Morel-Desrosiers.

**Data curation:** Milka Popova, Nicole Morel-Desrosiers.

**Formal analysis:** Rafael Muñoz-Tamayo, Milka Popova, Nicole Morel-Desrosiers.

**Funding acquisition:** Rafael Muñoz-Tamayo, Milka Popova.

**Investigation:** Rafael Muñoz-Tamayo, Milka Popova, Maxence Tillier, Nicole Morel-Desrosiers.

**Methodology:** Rafael Muñoz-Tamayo, Milka Popova, Maxence Tillier, Diego P. Morgavi, Jean-Pierre Morel, Gérard Fonty, Nicole Morel-Desrosiers.

**Software:** Rafael Muñoz-Tamayo.

**Writing – original draft:** Rafael Muñoz-Tamayo, Milka Popova, Nicole Morel-Desrosiers.

**Writing – review & editing:** Rafael Muñoz-Tamayo, Milka Popova, Diego P. Morgavi, Jean-Pierre Morel, Gérard Fonty, Nicole Morel-Desrosiers.

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
