## [Decision Letter · Decision Letter 0]

26 Sep 2019

PONE-D-19-19604

Hydrogenotrophic methanogens of the mammalian gut: functionally similar, thermodynamically different - A modelling approach

PLOS ONE

Dear Dr. Muñoz-Tamayo,

Thank you for submitting your manuscript to PLOS ONE. After careful consideration, we feel that it has merit but does not fully meet PLOS ONE’s publication criteria as it currently stands. Therefore, we invite you to submit a revised version of the manuscript that addresses the points raised during the review process.

We would appreciate receiving your revised manuscript by Nov 10 2019 11:59PM. To enhance the reproducibility of your results, we recommend that if applicable you deposit your laboratory protocols in protocols.io, where a protocol can be assigned its own identifier (DOI) such that it can be cited independently in the future. For instructions see: http://journals.plos.org/plosone/s/submission-guidelines#loc-laboratory-protocols

We look forward to receiving your revised manuscript.

Kind regards,

James E. Wells, PhD

Academic Editor

PLOS ONE

Journal Requirements:

Reviewers' comments:

Reviewer's Responses to Questions

**Comments to the Author**

1. Is the manuscript technically sound, and do the data support the conclusions?

Reviewer #1: Yes

Reviewer #2: Yes

2. Has the statistical analysis been performed appropriately and rigorously? 

Reviewer #1: Yes

Reviewer #2: N/A

3. Have the authors made all data underlying the findings in their manuscript fully available?

Reviewer #1: Yes

Reviewer #2: No

4. Is the manuscript presented in an intelligible fashion and written in standard English?

Reviewer #1: Yes

Reviewer #2: Yes

5. Review Comments to the Author

Reviewer #1: Manuscript Number: PONE-D-19-19604

Full Title: Hydrogenotrophic methanogens of the mammalian gut: functionally similar,

thermodynamically different - A modelling approach

General Comments:

The manuscript describes construction of a computer modelling analysis of the growth of three methanogenic archaea. A calorimetric approach was used along with more standard growth measurements. The manuscript was clearly written and logically organized. The growth conditions were quite standard in vitro batch cultures, but the measurements were inventive and the analyses thorough. There are caveats to extrapolating conclusions from batch growth to the continuous, or fed-batch, type of growth in the animal gut. However, the conclusions do not cross that line. The primary conclusion is that the model can be used to inform other models, which is quite reasonable. The vast majority of studies currently published on the gut microbiota are just 16S microbiome sequences. It is refreshing to see another approach. The specific comments below are minor in nature. I hope that they are useful.

Specific Comments:

L33: “We suggest that ecological models of gut ecosystems require the integration of microbial kinetics with nonlinear behaviours related to spatial and temporal variations taking place in mammalian guts.” That makes sense.

L93: Please provide the chain of custody for each archaeon. Were each of them acquired directly from the DSM or were they maintained by other investigators or culture collections?

L101: What type of tubes were used? Was the pressure in the tubes 2.5 Pa at the start of the growth experiment? Please include liquid and gas headspace volumes. These types of appropriate details are included in the Microcalorimetry section.

L107: Briefly describe the gas chromatography method.

L142: Is this the media described in the supplemental table? If so, define it as Balch growth media in the media section of the materials and methods.

L202: It seems reasonable that H2 would be the limiting substrate due to poor solubility, as mentioned on L182. That is the reason that the headspace is initially 80% H2. I wonder how the rate of diffusion into the liquid phase compares to the rate of H2 production by bacteria and other microbiota in vivo. Please comment here or elsewhere.

L222: Do the physiologies of the organisms support the assumption that ammonia is the only nitrogen source? Do they obligately synthesize all of their amino acids?

L267: “Tackling” is a colloquialism. “Before initiating the numerical estimation…” or “Before numerically estimating…”

L291: Elsewhere the manuscript states that log-phase cultures were used. The cultures were frozen at -20 degrees C. Three hours post inoculation there was about 50% variation in the viable number among the three species. That seems reasonable. There is no question or correction here. The reviewer notes that there were no problems with cell viability at the start of the experiment.

L303, L308 and elsewhere – The Results reads like a combined Results and Discussion section. It is recommended that interpretation be reserved for the Discussion section.

L358 – Check the markers in the figure legend. The manuscript handling system might have translated them incorrectly when the manuscript was changed to a PDF.

L370 – What are the black arrows on the figure panels? Please expand in the figure legend.

L388 – Based on figure 4, it looks like maximum specific growth rates were observed. That is, they look like ordinary batch growth curves in substrate-excess conditions. If the maximum specific growth rate for each organism was observed, then the initial H2 concentration must not have been limiting. Please discuss this point.

L400 – This is interesting discussion. In saccharolytic bacteria, the rate limiting factor is sometimes the rate substrate transport across the membrane. Once again, however, if the substrate was truly limiting, then the methanogens did not achieve their maximum specific growth rates. If the substrate were a sugar, I would want to know the sugar concentration throughout the curve. The diffusion of the H2 makes the question more complicated.

Reviewer #2: Authors assessed archaeal growth with cell count analysis based on OD660 and qPCR analysis, and measured gas pressure and composition. In addition, microcalorimetric measurements were performed for quantifying enthalpy, entropy and Gibbs energy change. Furthermore, in vitro and in silico mathematical modeling was applied to simulate dynamics of archaeal growth.

In the first paragraph of the introduction, which lacks a clear structure that smoothly narrows down to the objective of the paper, the authors mention gut archaea in relation to: 1) energy balance of the host 2) immune system of the host 3) methane as a terminal electron acceptor 4) methane as a greenhouse gas 5) cytochromes that they do or do not contain. In the second paragraph the authors report thermodynamics to be an important concept for dynamically predicting metabolic dynamics in the gut and state despite previously developed modeling frameworks, new knowledge could improve the predictive accuracy of these frameworks. It is not mentioned in the paper to what extent these previously developed models are inaccurate and what specific aspects of these models require improvement. The authors then aim to quantitatively characterize metabolic dynamics of three hydrogenotrophic gut methanogens. I would like the authors to state very clearly what problem they intend to tackle with the present work and to identify shortcomings of published studies. Also, it is insufficiently clear why the authors performed the microcalorimetry. What was done by the authors was already known for M. thermoautotrophicum, which employs exactly the same hydrogenotrophic methanogenic reaction.

It might be very valid that a model for three methanogenic species is defined in Eq 14 and that in the discussion of the paper the coexistence of the three methanogenic species is discussed. Please clarify why this is in line with the aim of the paper. The discussion regarding species coexistence is very interesting, but it is questionable if the classical competitive exclusion principle does actually not apply. Is the approach given by Eq 14 fully accurate? Archaea may be subject to different passage rates (values of ‘b’) for various reasons. They may transition back and forth between fluid and particulate matter, or either or not adhere to protozoa, live in syntrophy etc.

Is ‘energetic and kinetic differences between methanogens’ a proper name for the first subsection of the discussion? In the literature, energetic often refers to thermodynamics, which is in the name of the next subsection.

The authors refer to the gas phase and liquid phase of the rumen throughout the paper. Would it not be more clear to refer to liquid fraction and gas layer, because phase commonly refers to the state of a chemical substance? For example, carbon dioxide may transition from the gas to fluid phase a very low temperatures. In addition, hydrogen, carbon dioxide and methane are not in the liquid phase, but dissolved in aqueous solution.

It is somewhat difficult to understand how the reader should interpret Fig 2. Could you please revise the manuscript text such that most readers will interpret this figure as it should be interpreted?

I suspect the results from the in vitro work were used for parameter estimation of the model(s), but this is not stated in the paper. Could you please make this explicit?

The reported CCC and R^2 in Table 3 seem unrealistically high. Were the model predictions evaluated independently? If not, what is the value of this model evaluation?

6. PLOS authors have the option to publish the peer review history of their article (what does this mean?). If published, this will include your full peer review and any attached files.

Reviewer #1: No

Reviewer #2: No

---

## [Author Response · Author response to Decision Letter 0]

8 Oct 2019

Dear Dr. James E. Wells Academic Editor PlOS ONE, 

We would like to thank the reviewers for their assessment of our work and their constructive comments. 

We have decided to submit a revised version of the article. In the following, we address the comments of each reviewer.

We believe that the current version improves in clarity. We hope that you will find the revised version suitable for publication in PLOS ONE. In the modified version, we provided the links to get access to the code of the model and the experimental data. 

Sincerely 

On behalf of the authors 

Rafael Muñoz-Tamayo

Reviewer #1: 

General Comments:

 The manuscript describes construction of a computer modelling analysis of the growth of three methanogenic archaea. A calorimetric approach was used along with more standard growth measurements. The manuscript was clearly written and logically organized. The growth conditions were quite standard in vitro batch cultures, but the measurements were inventive and the analyses thorough. There are caveats to extrapolating conclusions from batch growth to the continuous, or fed-batch, type of growth in the animal gut. However, the conclusions do not cross that line. The primary conclusion is that the model can be used to inform other models, which is quite reasonable. The vast majority of studies currently published on the gut microbiota are just 16S microbiome sequences. It is refreshing to see another approach. The specific comments below are minor in nature. I hope that they are useful. 

Response: we thank the reviewer for her/his assessment and the usefulness of the comments 

Specific Comments:

L33: “We suggest that ecological models of gut ecosystems require the integration of microbial kinetics with nonlinear behaviours related to spatial and temporal variations taking place in mammalian guts.” That makes sense.

Response: thanks.

L93: Please provide the chain of custody for each archaeon. Were each of them acquired directly from the DSM or were they maintained by other investigators or culture collections?

Response: we added the requested information, lines 91-95. 

L101: What type of tubes were used? Was the pressure in the tubes 2.5 Pa at the start of the growth experiment? Please include liquid and gas headspace volumes. These types of appropriate details are included in the Microcalorimetry section.

Response: the tubes are now described on line 95. The initial pressure is shown in Supplementary Table 2. For the liquid and gas headspace volumes, the information was added in lines 94-95. Thank you for pointing out this missing information. 

L107: Briefly describe the gas chromatography method.

Response: done, lines 106-109

L142: Is this the media described in the supplemental table? If so, define it as Balch growth media in the media section of the materials and methods.

Response: done, line 93

L202: It seems reasonable that H2 would be the limiting substrate due to poor solubility, as mentioned on L182. That is the reason that the headspace is initially 80% H2. I wonder how the rate of diffusion into the liquid phase compares to the rate of H2 production by bacteria and other microbiota in vivo. Please comment here or elsewhere.

Response: this is a very relevant comment. We do not know of any studies reporting data on the rate of diffusion in the liquid phase in vivo. In the gut, after its production by fermenting microbes, hydrogen diffuses through the cell membrane in a dissolved form. The diffusion rate is dependent on the cell physiology (size and form), but also on hydrogen concentration in the cell’s environment. It has been suggested that dissolved H2 is supersaturated indicating lack of equilibrium between gas and liquid phases (1). The dissolved H2 concentration is about 0.1–50 µM. Dedicated experiments are needed to quantify the diffusion rate under in vivo conditions where H2 production is also influenced by interspecies H2 transfer from bacteria and protozoa to hydrogenotrophic archaea. 

L222: Do the physiologies of the organisms support the assumption that ammonia is the only nitrogen source? Do they obligately synthesize all of their amino acids?

Response: the reviewer is right in pointing out that microbes can utilize different sources of nitrogen. We used the simplified assumption that NH3 is the sole nitrogen source, following the model developments in anaerobic digestion (4) and the thermodynamic study on Methanobacterium thermoautotrophicum (5). 

L267: “Tackling” is a colloquialism. “Before initiating the numerical estimation…” or “Before numerically estimating…”

Response: corrected L267

L291: Elsewhere the manuscript states that log-phase cultures were used. The cultures were frozen at -20 degrees C. Three hours post inoculation there was about 50% variation in the viable number among the three species. That seems reasonable. There is no question or correction here. The reviewer notes that there were no problems with cell viability at the start of the experiment.

Response: Thank you for your comment, really appreciated

L303, L308 and elsewhere – The Results reads like a combined Results and Discussion section. It is recommended that interpretation be reserved for the Discussion section.

Response: we follow the advice of the reviewer L389-394.

L358 – Check the markers in the figure legend. The manuscript handling system might have translated them incorrectly when the manuscript was changed to a PDF.

Response: the markers are correct 

L370 – What are the black arrows on the figure panels? Please expand in the figure legend.

Response: the arrows were put to indicate the unit axis of each biomass. We realized the arrows are unnecessary so we deleted them from the figure.

L388 – Based on figure 4, it looks like maximum specific growth rates were observed. That is, they look like ordinary batch growth curves in substrate-excess conditions. If the maximum specific growth rate for each organism was observed, then the initial H2 concentration must not have been limiting. Please discuss this point.

Response: the reviewer is right. In the experiments we overpressed tubes with hydrogen to ensure substrate availability in excess. We used wrongly the term limiting substrate. What we wanted to express is that we used a single substrate kinetic rate that is function of the hydrogen concentration only (rather than a kinetic rate equation with two substrates). Corrections were done L204, 384.

L400 – This is interesting discussion. In saccharolytic bacteria, the rate limiting factor is sometimes the rate substrate transport across the membrane. Once again, however, if the substrate was truly limiting, then the methanogens did not achieve their maximum specific growth rates. If the substrate were a sugar, I would want to know the sugar concentration throughout the curve. The diffusion of the H2 makes the question more complicated.

Response: in the previous response we confirmed the observation of the reviewer that our conditions were not under substrate limitation. It is true that the diffusion of H2 complicates the analysis. Given the closeness between the measured total heat and the expected theoretical value calculated from the methanogenesis reaction, we might think that diffusion process does not contribute significantly to the heat of the complete process. However, further studies will be required to provide evidence.

Reviewer #2: 

Authors assessed archaeal growth with cell count analysis based on OD660 and qPCR analysis, and measured gas pressure and composition. In addition, microcalorimetric measurements were performed for quantifying enthalpy, entropy and Gibbs energy change. Furthermore, in vitro and in silico mathematical modeling was applied to simulate dynamics of archaeal growth.

In the first paragraph of the introduction, which lacks a clear structure that smoothly narrows down to the objective of the paper, the authors mention gut archaea in relation to: 1) energy balance of the host 2) immune system of the host 3) methane as a terminal electron acceptor 4) methane as a greenhouse gas 5) cytochromes that they do or do not contain. In the second paragraph the authors report thermodynamics to be an important concept for dynamically predicting metabolic dynamics in the gut and state despite previously developed modeling frameworks, new knowledge could improve the predictive accuracy of these frameworks. It is not mentioned in the paper to what extent these previously developed models are inaccurate and what specific aspects of these models require improvement. The authors then aim to quantitatively characterize metabolic dynamics of three hydrogenotrophic gut methanogens. I would like the authors to state very clearly what problem they intend to tackle with the present work and to identify shortcomings of published studies. Also, it is insufficiently clear why the authors performed the microcalorimetry. What was done by the authors was already known for M. thermoautotrophicum, which employs exactly the same hydrogenotrophic methanogenic reaction.

Response: the Introduction was modified to improve clarity L61-86. We stated clearly the objective of our work. Microcalorimetric experiments were performed to identify differences in the dynamic energetic pattern of the three methanogens. These experiments were instrumental to estimate the specific growth rates of the microbes. 

It might be very valid that a model for three methanogenic species is defined in Eq 14 and that in the discussion of the paper the coexistence of the three methanogenic species is discussed. Please clarify why this is in line with the aim of the paper. The discussion regarding species coexistence is very interesting, but it is questionable if the classical competitive exclusion principle does actually not apply. Is the approach given by Eq 14 fully accurate? Archaea may be subject to different passage rates (values of ‘b’) for various reasons. They may transition back and forth between fluid and particulate matter, or either or not adhere to protozoa, live in syntrophy etc.

Response: The interest of addressing the competition exclusion principle is defined in the Introduction section L80-83

We acknowledge that our model is a very simplified representation. It is why we used the term “toy model” to express the model limitation. The remarks of the reviewer about the different passages of the microbes (values of Di) and variable output substrate rate (parameter b) align with our conclusion that the classical competitive exclusion principle does not apply, since the original form of the exclusion principle considers a constant output substrate rate and the same passage rate for the microbes. A theoretical development will be needed to explain methanogens coexistence in the gut.

Is ‘energetic and kinetic differences between methanogens’ a proper name for the first subsection of the discussion? In the literature, energetic often refers to thermodynamics, which is in the name of the next subsection.

Response: we changed the name of the second subsection to be consistent L413

The authors refer to the gas phase and liquid phase of the rumen throughout the paper. Would it not be more clear to refer to liquid fraction and gas layer, because phase commonly refers to the state of a chemical substance? For example, carbon dioxide may transition from the gas to fluid phase a very low temperatures. In addition, hydrogen, carbon dioxide and methane are not in the liquid phase, but dissolved in aqueous solution.

Response: the reviewer is right. However, we prefer to use the term phase because it is widely used in the literature of anaerobic digestion modelling that inspired our model developments (2,4). 

It is somewhat difficult to understand how the reader should interpret Fig 2. Could you please revise the manuscript text such that most readers will interpret this figure as it should be interpreted?

Response: the Figure and the legend were modified L311.

I suspect the results from the in vitro work were used for parameter estimation of the model(s), but this is not stated in the paper. Could you please make this explicit?

Response: the information was added in L285-286

The reported CCC and R^2 in Table 3 seem unrealistically high. Were the model predictions evaluated independently? If not, what is the value of this model evaluation?

Response: if we understood correctly, the reviewer suggests to report the statistics of model performance for each methanogen. We followed this advice and modified the Table 3 accordingly as well as the values in the manuscript L23,24, L356-357. In addition, we made the data available L352-353. The reviewer can check our calculations. 

References

1. Wang M, Ungerfeld EM, Wang R, Zhou CS, Basang ZZ, Ao SM, et al. Supersaturation of dissolved hydrogen and methane in rumen of Tibetan sheep. Front Microbiol. 2016; 

2. Janssen PH. Influence of hydrogen on rumen methane formation and fermentation balances through microbial growth kinetics and fermentation thermodynamics. Anim Feed Sci Technol. 2010;160:1–22. 

3. Muñoz-Tamayo R, Giger-Reverdin S, Sauvant D. Mechanistic modelling of in vitro fermentation and methane production by rumen microbiota. Anim Feed Sci Technol. 2016;220:1–21. 

4. Batstone DJ, Keller J, Angelidaki I, Kalyuzhnyi S V, Pavlostathis SG, Rozzi A, et al. Anaerobic Digestion Model No.1 (ADM1). IWA Task Group for Mathematical Modelling of Anaerobic Digestion Processes. IWA Publishing, London; 2002. 

5. Schill NA, Liu JS, von Stockar U. Thermodynamic analysis of growth of Methanobacterium thermoautotrophicum. Biotechnol Bioeng. 1999;64:74–81.

---

## [Decision Letter · Decision Letter 1]

18 Nov 2019

PONE-D-19-19604R1

Hydrogenotrophic methanogens of the mammalian gut: functionally similar, thermodynamically different - A modelling approach

PLOS ONE

Dear Dr. Muñoz-Tamayo,

Thank you for submitting your manuscript to PLOS ONE. After careful consideration, we feel that it has merit but does not fully meet PLOS ONE’s publication criteria as it currently stands. Therefore, we invite you to submit a revised version of the manuscript that addresses the points raised during the review process.

We would appreciate receiving your revised manuscript by Jan 02 2020 11:59PM. To enhance the reproducibility of your results, we recommend that if applicable you deposit your laboratory protocols in protocols.io, where a protocol can be assigned its own identifier (DOI) such that it can be cited independently in the future. For instructions see: http://journals.plos.org/plosone/s/submission-guidelines#loc-laboratory-protocols

We look forward to receiving your revised manuscript.

Kind regards,

James E. Wells, PhD

Academic Editor

PLOS ONE

Additional Editor Comments (if provided):

Reviewer 2 has one concern that needs to be addressed.

Reviewers' comments:

Reviewer's Responses to Questions

**Comments to the Author**

1. If the authors have adequately addressed your comments raised in a previous round of review and you feel that this manuscript is now acceptable for publication, you may indicate that here to bypass the “Comments to the Author” section, enter your conflict of interest statement in the “Confidential to Editor” section, and submit your "Accept" recommendation.

Reviewer #1: All comments have been addressed

Reviewer #2: (No Response)

2. Is the manuscript technically sound, and do the data support the conclusions?

Reviewer #1: Yes

Reviewer #2: Yes

3. Has the statistical analysis been performed appropriately and rigorously? 

Reviewer #1: Yes

Reviewer #2: No

4. Have the authors made all data underlying the findings in their manuscript fully available?

Reviewer #1: Yes

Reviewer #2: Yes

5. Is the manuscript presented in an intelligible fashion and written in standard English?

Reviewer #1: Yes

Reviewer #2: Yes

6. Review Comments to the Author

Reviewer #1: (No Response)

Reviewer #2: I appreciate the rewrite of the introduction by the authors, which definitely made the paper improve in clarity. However, I feel the authors can still make the very final step regarding the added value of the paper to the existing literature. I suggest that the authors add a last piece of information to the discussion section to discuss why the present model development would contribute to increased understanding of the immune system of humans and the prediction of greenhouse gases from the rumen. Or how suggested further model development dealing with nonlinear behaviour related to spatial and temporal variation in mammalian gut systems contributes to this? Do we need to be able to model coexistence to answer these questions?

I apologise for unclarity from my side regarding the reported CCC and R^2 in Table 3 that seemed unrealistically high. What I meant is that it appears to me that the model parameters were fitted to the data that was available. Using those fitted parameters, predicted values were obtained from model simulations and compared with observed values. This comparison resulted in CCC and R^2 values. My point is that the same data should not be used for model parameter fitting and model evaluation to ensure that model predictions are evaluated independently. If a model is not evaluated independently, observed vs. predicted plots could serve as valid diagnostic plots, but CCC and R^2 values are trivial. Independent data is needed for a solid model evaluation. The authors may split the data that is available if they are able to do so.

I wonder why the x-axes of Figures 3 and 4 run over 75 and 100 h, respectively, instead of 75 h for both Figures.

7. PLOS authors have the option to publish the peer review history of their article (what does this mean?). If published, this will include your full peer review and any attached files.

Reviewer #1: No

Reviewer #2: No

---

## [Author Response · Author response to Decision Letter 1]

21 Nov 2019

Dear Dr. James E. Wells Academic Editor PlOS ONE, 

We have decided to submit a revised version of the article. Below, we addressed the comments of reviewer 2.

We hope that you will find the revised version suitable for publication in PLOS ONE. 

Sincerely 

On behalf of the authors 

Rafael Muñoz-Tamayo

Reviewer #2: 

I appreciate the rewrite of the introduction by the authors, which definitely made the paper improve in clarity. However, I feel the authors can still make the very final step regarding the added value of the paper to the existing literature. I suggest that the authors add a last piece of information to the discussion section to discuss why the present model development would contribute to increased understanding of the immune system of humans and the prediction of greenhouse gases from the rumen. Or how suggested further model development dealing with nonlinear behaviour related to spatial and temporal variation in mammalian gut systems contributes to this? Do we need to be able to model coexistence to answer these questions?

Response: characterising the dynamics of rumen methanogens (and other rumen microbes) is essential to the enhanced understanding of the microbial community functioning and thus holobiont’s phenotype. In particular, the efficiency of methane mitigation strategies are strongly dependent of dynamic properties such as the rumen ecosystem resilience and also by the functional redundancy of microbes (Weimer, 2015). This functional redundancy is related to the aspects discussed in our article. Our modelling work aimed to characterise methanogen dynamics. We prefer in the paper to avoid being speculative about the impact of our model on the prediction of in vivo systems and focus on our actual outcomes avoiding the risk of overselling our findings. On-going and future work will aim at expanding the model to the gut ecosystem, but we are aware that we have a long way. 

In regard to the relevance of incorporating nonlinear behaviour related to spatial and temporal variation in mammalian ecosystems, these aspects have been already discussed in our previous developments (Muñoz-Tamayo et al., 2010) and by other authors (Widder et al., 2016). Spatio-temporal mechanisms are determining for the functioning and colonization of the human gut (Labarthe et al., 2019). An experimental work indicated the importance of temporal dynamics to enhance the understanding on rumen function (Huws et al., 2016).

The level of detail of the model is an open question for the modeller and depends of the question the model is intended to help to answer. For example, as demonstrated in our modelling works, we do not need information of the rumen microbiota to predict accurately enteric methane production (Muñoz-Tamayo et al., 2019). However, microbial information is needed into models to inform on manipulation strategies and improving understanding of rumen microbial function as discussed in our recent review (Huws et al., 2018). For the human gut, analysing competition and syntrophy between gut microbes allows to identify assembling rules of the microbial community. “Notably, elucidating the assembly rules of the microbiome goes beyond gaining a better understanding of basic ecological processes and has profound clinical implications.”(Levy and Borenstein, 2013)

These interesting questions should be addressed integrating new models and data from those used in our work. 

I apologise for unclarity from my side regarding the reported CCC and R^2 in Table 3 that seemed unrealistically high. What I meant is that it appears to me that the model parameters were fitted to the data that was available. Using those fitted parameters, predicted values were obtained from model simulations and compared with observed values. This comparison resulted in CCC and R^2 values. My point is that the same data should not be used for model parameter fitting and model evaluation to ensure that model predictions are evaluated independently. If a model is not evaluated independently, observed vs. predicted plots could serve as valid diagnostic plots, but CCC and R^2 values are trivial. Independent data is needed for a solid model evaluation. The authors may split the data that is available if they are able to do so.

Response: we recognize the importance of model validation (falsification). However, model validation is not an exclusive criterion for model evaluation. The lack of the model validation step does not preclude a rigour analysis of model evaluation. The implementation of a model validation requires a large data set. In our study, the number of sampling times for each microbe is limited. Accordingly, splitting the data for calibration and validation is not a good strategy. Reducing the number of data for model calibration will reduce the informative content of the process dynamics and thus will have a detrimental effect in the accuracy of the parameter estimates. Model validation is desired when data is available under other experimental conditions (e.g. operation under continuous mode), which is not the case in our study. We disagree with the analysis the reviewer performs with respect to the statistic indicators for model evaluation. The values of R2 and CCC calculated under the calibration context are not trivial since a model with structural problems will lead to unsatisfactory R2 and CCC. The high values that we obtained for R2 and CCC demonstrated that the model is well structured and that captures the dynamic of the methanogenesis. Our satisfactory calibration outcomes are the result of an adequate model construction that includes the theoretical identifiability property of the model as discussed in the article and also from the fact that we addressed practical identifiability issues. The sampling times were actually determined using an optimal experiment design (OED) strategy to facilitate accurate estimation of the parameters. This OED strategy was based using estimates from preliminary data and implementing an optimization problem that maximizes the determinant of the Fisher Information Matrix (Muñoz-Tamayo et al., 2014). This information is not detailed in the article to avoid the reader to get lost in mathematical technicalities. It should be said that our calibration strategy follows a validation-like principle since the maximum specific growth rate constants were obtained from the calorimetric experiments data and injected further into the dynamic model challenged with the data from the growth kinetics experiments. This procedure strengthens the quality of our model.

In lines 410-413, we recognized the limitation of our work with respect to model validation.

I wonder why the x-axes of Figures 3 and 4 run over 75 and 100 h, respectively, instead of 75 h for both Figures.

Response: Figure 4 was modified to get the axes homogenous 

References 

Huws, S.A., Creevey, C.J., Oyama, L.B., Mizrahi, I., Denman, S.E., Popova, M., et al. (2018) Addressing global ruminant agricultural challenges through understanding the rumen microbiome: past, present, and future. Front Microbiol 9: 2161.

Huws, S.A., Edwards, J.E., Creevey, C.J., Stevens, P.R., Lin, W., Girdwood, S.E., et al. (2016) Temporal dynamics of the metabolically active rumen bacteria colonizing fresh perennial ryegrass. FEMS Microbiol Ecol 92:.

Labarthe, S., Polizzi, B., Phan, T., Goudon, T., Ribot, M., and Laroche, B. (2019) A mathematical model to investigate the key drivers of the biogeography of the colon microbiota. J Theor Biol 462: 552–581.

Levy, R. and Borenstein, E. (2013) Metabolic modeling of species interaction in the human microbiome elucidates community-level assembly rules. Proceeding Natl Acad Sci United States Am 110: 12804–12809.

Muñoz-Tamayo, R., Laroche, B., Walter, E., Doré, J., and Leclerc, M. (2010) Mathematical modelling of carbohydrate degradation by human colonic microbiota. J Theor Biol 266: 189–201.

Muñoz-Tamayo, R., Martinon, P., Bougaran, G., Mairet, F., and Bernard, O. (2014) Getting the most out of it: Optimal experiments for parameter estimation of microalgae growth models. J Process Control 24:.

Muñoz-Tamayo, R., Ramírez Agudelo, J.F., Dewhurst, R.J., Miller, G., Vernon, T., and Kettle, H. (2019) A parsimonious software sensor for estimating the individual dynamic pattern of methane emissions from cattle. Animal 13: 1180–1187.

Weimer, P.J. (2015) Redundancy, resilience, and host specificity of the ruminal microbiota: implications for engineering improved ruminal fermentations. Front Microbiol 6: 296.

Widder, S., Allen, R.J., Pfeiffer, T., Curtis, T.P., Wiuf, C., Sloan, W.T., et al. (2016) Challenges in microbial ecology: Building predictive understanding of community function and dynamics. ISME J 10: 2557–2568.

---

## [Editor Report · Decision Letter 2]

25 Nov 2019

Hydrogenotrophic methanogens of the mammalian gut: functionally similar, thermodynamically different - A modelling approach

PONE-D-19-19604R2

Dear Dr. Muñoz-Tamayo,

We are pleased to inform you that your manuscript has been judged scientifically suitable for publication and will be formally accepted for publication once it complies with all outstanding technical requirements.

With kind regards,

James E. Wells, PhD

Academic Editor

PLOS ONE
---

## [Editor Report · Acceptance letter]

3 Dec 2019

PONE-D-19-19604R2 

Hydrogenotrophic methanogens of the mammalian gut: functionally similar, thermodynamically different - A modelling approach 

Dear Dr. Muñoz-Tamayo:

I am pleased to inform you that your manuscript has been deemed suitable for publication in PLOS ONE. Congratulations! Your manuscript is now with our production department. 

With kind regards,

on behalf of

Dr. James E. Wells 

Academic Editor

PLOS ONE